# In Vitro Microleakage Comparison of Three Pit-and-Fissure Sealant Protocols: Self-Etch Sealant with and Without Separate Adhesive Versus Conventional Resin-Based Sealants

**DOI:** 10.3390/biomedicines13122902

**Published:** 2025-11-27

**Authors:** Catalina Iulia Saveanu, Alexandra Sestac, Daniela Anistoroaei, Alexandra Ecaterina Saveanu, Mioara Florentina Trandafirescu, Maria Sophia Saveanu, Loredana Golovcencu

**Affiliations:** 1Department of Surgical, Faculty of Dental Medicine, Grigore T Popa University of Medicine and Pharmacy, 700115 Iasi, Romania; catalina.saveanu@umfiasi.ro (C.I.S.); ale_alexandra20079@yahoo.com (A.S.); saveanu.alexandra-ecaterina@d.umfiasi.ro (A.E.S.); md-rom-10944@students.umfiasi.ro (M.S.S.); loredana.golovcencu@umfiasi.ro (L.G.); 2Department of Morphofunctional Sciences I—Histology, Faculty of Medicine, Grigore T Popa University of Medicine and Pharmacy, 700115 Iasi, Romania; mio.trandafirescu@umfiasi.ro

**Keywords:** pit-and-fissure sealants, microleakage, adhesive systems, sealing cracks, in vitro, resin-based materials, dye penetration, enamel bonding

## Abstract

**Background:** Pit-and-fissure sealants are a fundamental preventive strategy in occlusal caries management, particularly in children and adolescents. The effectiveness of sealants largely depends on their marginal sealing ability, which prevents microleakage and subsequent bacterial penetration. **Objective:** This in vitro study aimed to compare the microleakage performance of three sealant protocols: Fissurit FX (VOCO GmbH, Cuxhaven, Germany), Perma Seal Perma Seal (Ultradent Products Inc., South Jordan, UT, USA) with an adhesive system, and Perma Seal without adhesive. **Methods:** A total of 18 extracted human molars were randomly divided into three groups (n = 6/group). Following standard enamel cleaning and acid etching, sealants were applied according to the manufacturer’s instructions. Samples were thermocycled, immersed in 5% methylene blue, sectioned bucco-lingually, and evaluated for dye penetration under magnification. Microleakage was scored on a 0–3 scale. Intra-examiner reliability was assessed with Cohen’s kappa. Statistical analysis included Kruskal–Wallis, Chi-Square, and Wilcoxon signed-rank tests. **Results:** The Kruskal–Wallis test revealed significant differences between groups (*p* = 0.006). Perma Seal with adhesive demonstrated the lowest microleakage scores, followed by Fissurit FX, while Perma Seal without adhesive showed the highest leakage. Cohen’s kappa of 0.952 indicated excellent scoring reliability. Chi-Square analysis confirmed the association between material and leakage pattern (*p* = 0.003). **Conclusions:** The use of an adhesive system prior to self-etch sealant application significantly enhances marginal sealing and reduces microleakage. These findings support the incorporation of adhesive pre-treatment protocols into clinical practice to optimize sealant efficacy.

## 1. Introduction

Dental caries is a prevalent chronic disease with a multifactorial etiology, influenced by biological, environmental, and behavioral factors. It results from an ecological imbalance within the oral biofilm, favoring acidogenic and aciduric bacteria, which leads to net mineral loss in the dental hard tissues [1]. The earliest clinical sign of enamel demineralization is the appearance of a white-spot lesion, which is most frequently observed on the occlusal surfaces of posterior teeth—particularly in pits and fissures, where plaque retention is facilitated by the complex anatomical morphology.

Despite the global decline in smooth-surface caries attributable to fluoride exposure, occlusal caries remains predominant in school-aged children. Recent data confirm that untreated carious lesions in permanent teeth remain one of the most prevalent health conditions globally, affecting over a billion people worldwide [2]. Consequently, occlusal surfaces—especially of first and second molars—are still the principal sites of new lesions in young patients, due to their anatomical complexity and limited access to brushing and fluoride [3].

Conventional resin-based sealants, typically Bis GMA-based, require multistep application including etching, adhesive priming, and light curing; the technique is effective but highly operator- and moisture-sensitive, especially in pediatric settings [4].

Self-etch composite sealants have emerged to reduce procedural complexity. These sealants incorporate acidic monomers capable of conditioning enamel and forming micromechanical bonds without separate acid-etching or rinsing—potentially reducing application error, chair time, and improving performance under suboptimal isolation [5].

Clinical evidence indicates that retention rates for self-etch sealants are comparable to conventional protocols, particularly when a hydrophobic primer is used [6]. However, other studies suggest that without selective pre-etching, especially on immature enamel of recently erupted molars, long term retention and marginal integrity may be compromised [6].

Microscopic and clinical evidence further supports the role of enamel surface conditioning in improving sealant performance. Furthermore, a clinical follow-up study evaluating the performance of a Bis-GMA-based sealant, Defense-Chroma (Angelus Indústria de Produtos Odontológicos S/A, Londrina, Brazil) demonstrated a 94% full retention rate after six months when applied to newly erupted permanent molars under controlled conditions. In parallel, another study in vitro examined microleakage at the resin composite–enamel interface, highlighting that sealant performance is strongly influenced by factors such as material flowability, filler content, and the effectiveness of acid-etch conditioning in enhancing marginal adaptation and seal integrity [7].

These data underscore the importance of optimizing the enamel surface prior to sealant application, even when using self-etch materials, to maximize retention and minimize microleakage. This becomes particularly relevant in minimally invasive dentistry and preventive pediatric care, where non-invasive, efficient, and moisture-tolerant techniques are highly desirable [8].

Since self-etch sealants are sometimes applied directly onto etched enamel without an additional adhesive layer to simplify the procedure, we also included a Perma Seal (Ultradent Products Inc., South Jordan, UT, USA) group without adhesive as a comparator. This allowed us to evaluate whether omitting the bonding step influences marginal sealing and microleakage.

This in vitro study aims to evaluate and compare the degree of microleakage among three pit-and-fissure sealing protocols applied to occlusal surfaces of permanent molars: a conventional resin-based sealant (Fissurit FX (VOCO GmbH, Cuxhaven, Germany), a self-etch composite sealant Perma Seal (Ultradent Products Inc., South Jordan, UT, USA) applied with an adhesive system, and the same self-etch sealant applied without adhesive.

Study hypotheses:

**Null** **Hypothesis** **(H_0_).**
*There is no statistically significant difference in microleakage among the three sealing protocols: conventional resin-based sealant, self-etch sealant with adhesive, and self-etch sealant without adhesive.*


**Alternative** **Hypothesis** **(H_1_).**
*There is a statistically significant difference in microleakage among the three sealing protocols, with the use of an adhesive system expected to reduce leakage in self-etch sealants.*


## 2. Materials and Methods

### 2.1. Study Design and Ethical Considerations

This in vitro, randomized, controlled laboratory study, was conducted at the Grigore T. Popa University of Medicine and Pharmacy Iasi, following ISO/TS 11405:2015 standards [9] for testing adhesion to dental hard tissues. The study was approved by the Institutional Ethics Committee Approval No. 167/21.03.2022 and followed the principles of the Declaration of Helsinki. All teeth were obtained with informed consent and used in accordance with biohazard safety protocols.

### 2.2. Tooth Selection and Preparation

A total of 18 extracted permanent molars were collected from adult patients undergoing extractions for orthodontic or periodontal reasons. No distinction was made between male and female donors, as sex was not considered to influence the outcomes of this in vitro study. Only intact or non-cavitated teeth, free of restoration, cracks, or developmental defects, were included. All teeth were obtained with informed consent, and the study protocol received institutional ethical approval. After extraction, the teeth were disinfected in 3% hydrogen peroxide, cleaned manually, and polished using Depural Neo Depural Ne (Pentron/Spofa Dental, Jičín, Czech Republic) prophylactic paste and a rotary prophylaxis brush (3M ESPE, St. Paul, MN, USA). The cleaning paste was removed and dried using an air–water spray oil-free air (Dürr Dental, Bietigheim-Bissingen, Germany).

All pits and fissures were gently explored with a sharp probe to eliminate any residual pumice or debris. The initial sample selection consisted of extracted posterior teeth, both molars and premolars, meeting inclusion criteria for fissure sealant applications.

A total of 18 sound or non-cavitated posterior teeth (molars and premolars), extracted for orthodontic or periodontal reasons, were included in the study. The teeth exhibited occlusal anatomy suitable for sealant placement and no restorations or gross structural defects.

### 2.3. Sealant Materials

The teeth were randomly allocated into three experimental groups (n = 6 per group): Group 1: Fissurit FX sealant + Xbond adhesive system; Group 2: Perma Seal + Xbond adhesive system; Group 3: Perma Seal without adhesive system.

Table 1 provides a comprehensive overview of the clinical simulation protocols applied to each experimental group, highlighting the procedural differences related to sealant type and adhesive usage. All three groups followed a standardized sequence for enamel surface preparation, including prophylaxis, etching with 36% phosphoric acid gel, and drying to obtain the characteristic white enamel surface.

Group 1 received Fissurit FX sealant following the application of Xbond adhesive, representing a conventional approach combining etching and bonding prior to sealant placement. Group 2 used the same adhesive system but applied it in conjunction with PermaSeal, while for Group 3 the bonding step was eliminated, applying Perma Seal directly onto the etched enamel.

The inclusion of precise technical parameters for each material—including filler content, chemical composition, and manufacturer details—ensures full reproducibility of the experimental setup and allows comparison with other studies. Furthermore, the microleakage assessment method (dye penetration with methylene blue and sectioning technique) was uniformly applied across groups, enabling an objective evaluation of marginal integrity under controlled in vitro conditions.

This structured presentation emphasizes experimental consistency, while clearly delineating the independent variables under investigation: type of sealant and use of adhesive system.

Sealants were applied in preloaded syringes with fine applicator tips. The adhesive (Xbond) was a monocomponent system combining primer and bonding functions.

The LED curing unit used in this study emits a constant high-intensity blue light (420–480 nm; 1000–1600 mW/cm^2^), designed for rapid polymerization of photosensitive resins and whitening materials.

It features an enhanced fiber-optic light guide for improved convergence and uniform illumination. The device operates intermittently and is powered by a 3.7 V lithium-ion battery (2200 mAh). It has dimensions of 180 × 26 × 26 mm and supports input voltages of AC 100–240 V and DC 4.2 V/1 A. The unit is classified as a Class II, IP40 medical device and is equipped with safety fuses. A LED light-curing unit was used (LED.D, Guilin Woodpecker Medical Instrument Co., Guilin, China).

### 2.4. Sealant Application Protocol

The application of all sealants was performed in accordance with the protocols provided by the respective manufacturers. Table 2 summarizes the distinct procedural steps followed across the three experimental groups. All groups underwent identical surface conditioning, including phosphoric acid etching and standard air-drying to obtain a chalky white enamel surface. However, only Groups 1 and 2 utilized the Xbond bonding system, with subsequent air-drying and light-curing, while Group 3 received no adhesive application, relying solely on mechanical retention of the sealant.

While Fissurit FX was used exclusively in Group 1, Perma Seal was the material of choice in Groups 2 and 3, with the only variable being the presence or absence of the bonding agent. All samples underwent light-curing of the sealant and subsequent immersion in 5% methylene blue solution for microleakage evaluation, ensuring uniform conditions for dye penetration analysis.

This protocol matrix highlights the controlled comparison between adhesive-supported and non-adhesive protocols, as well as between two different sealant materials, providing a consistent framework for assessing the impact of bonding and material type on marginal sealing performance. All procedures were performed in a darkened environment to avoid premature polymerization.

To simulate the thermal variations that occur in the oral cavity, all samples were manually subjected to thermocycling, in accordance with the ISO 11405:2015 recommendations. The protocol consisted of 500 cycles, with alternate immersion of the specimens in two separate water baths maintained at 5 ± 2 °C and 55 ± 2 °C, respectively. Each immersion lasted 30 s, followed by a 10 s transfer time between baths. Temperature was regularly monitored using a calibrated digital thermometer to ensure accuracy. This method allowed for a reproducible simulation of thermal fatigue that dental materials may undergo clinically and is consistent with protocols used in previous in vitro microleakage studies.

After sealant application, each tooth was coated with two layers of nail varnish on both the crown and root surfaces, leaving a 1 mm margin around the sealed fissures exposed. The specimens were then immersed in 5% methylene blue dye solution (SD Industries, Mumbai, India) for 24 h at 37 °C to allow the dye to infiltrate any marginal gaps. After immersion, samples were rinsed with water for 30 s and air-dried.

Each tooth was then sectioned longitudinally in the bucco-lingual direction through the center of the sealed area using a high-speed tapered diamond bur under continuous water cooling. This procedure yielded two symmetrical halves per sample. A total of six teeth were processed for each of the three sealant groups, resulting in 12 half-samples per group.

### 2.5. Microleakage Assessment and Examiner Calibration

Microleakage was scored based on the Williams and Winters criteria [10].

Score 0 = No detectable dye penetration at the sealant–enamel interface. The dye is completely absent within the fissure, indicating excellent marginal sealing.

Score 1 = Dye penetration is limited to a superficial third of the fissure depth. The dye reaches only the upper part of the sealed groove, suggesting minimal microleakage.

Score 2 = Dye penetration extends from one-third to two-thirds of the fissure depth. Partial infiltration is evident, indicating moderate microleakage but not complete sealant failure.

Score 3 = Dye has penetrated beyond two-thirds of the fissure depth or reached the base of the fissure. This indicates substantial or complete failure of the marginal seal, with high risk of clinical leakage.

Microleakage was assessed separately on the buccal and oral surfaces of each half-section. Because each tooth provided two sectioned halves (labeled a and b), and each half was evaluated on two surfaces, four microleakage measurements were obtained per tooth.

In total, 24 microleakage scores were recorded for each experimental group, and 72 individual scores were analyzed overall. Sectioned surfaces were polished and examined under 7.2× magnification, and representative images were captured using a Samsung Galaxy S20 smartphone camera (12 MP; Samsung Electronics Co., Suwon, South Korea) to document dye penetration patterns. The complete scoring dataset is presented in Section 3.

### 2.6. Statistical Analysis

All statistical analysis were performed using IBM SPSS Statistics version 26.0 (IBM Corp., Armonk, NY, USA). The microleakage scores, ranging from 0 to 3, were treated as ordinal variables and analyzed using non-parametric tests, as the data did not meet the assumptions of normality. To compare the distribution of microleakage scores between the three experimental groups (Fissurit FX, Perma Seal with adhesive, and Perma Seal without adhesive), the Kruskal–Wallis H test was applied. A significance threshold of α = 0.05 was used. Effect size and power/sensitivity analysis. The primary omnibus comparison across the three sealant protocols was conducted with the Kruskal–Wallis test (k = 3 groups; unit of analysis = tooth; n = 18, n = 6 per group). For interpretability, we quantified the magnitude of the between-group effect using a rank-based effect size for Kruskal–Wallis, ηH^2^ = (H − k + 1)/(N − k). Based on the observed statistic (H = 10.243), the effect size was ηH^2^ ≈ 0.55 (large). For comparability with ANOVA-based metrics used in power analyses, we converted to Cohen’s f = η^2^/(1 − η^2^) ≈ 1.10. We also performed a sensitivity analysis using the standard one-way ANOVA (fixed effects, omnibus) approximation to Kruskal–Wallis (α = 0.05; power = 0.80; k = 3; n = 18), which indicated a minimal detectable effect of approximately f ≈ 0.81 (corresponding to η^2^ ≈ 0.40); under this approximation, the observed effect corresponds to an achieved power of ~0.99. Given the ordinal outcome, we report effect sizes and distributional summaries alongside *p*-values. Where applicable, pairwise comparisons were conducted using adjusted *p*-values (Bonferroni correction). To evaluate potential differences in microleakage between the buccal and oral surfaces within the same sample, the Wilcoxon signed-rank test was applied for each group separately (n = 12 paired observations per group). This allowed the assessment of whether sealant performance varied by surface orientation. Additionally, Cohen’s kappa coefficient was calculated to assess intra-examiner reliability for microleakage scoring. All 72 valid measurements were scored twice by the same calibrated examiner. The analysis yielded a kappa value of 0.952, indicating almost perfect agreement, according to the Landis and Koch interpretation scale. This result demonstrates a high degree of consistency and reproducibility in the visual scoring method used throughout the study.

## 3. Results

The present study aimed to evaluate and compare the microleakage performance of three pit-and-fissure sealant protocols, using an in vitro dye penetration model and a standardized scoring system. A total of 72 microleakage measurements were recorded across all experimental groups, with 24 scores per group (12 samples × 2 surfaces). Each sample was sectioned bucco-lingually and scored separately on the buccal (V) and oral (O) aspects.

To facilitate direct comparisons between surfaces and protocols, Table 3 reports the individual ordinal microleakage scores (0–3; higher values indicate more severe leakage) recorded on the buccal and oral surfaces for each specimen. Samples 1–6 correspond to Fissurit FX (F-FX), 7–12 to Perma Seal with adhesive system (PS + AS), and 13–18 to Perma Seal without adhesive system (PS-AS). For each tooth, (a) and (b) designate the two sectioned halves evaluated. The scoring criteria are described in Section 2.

### 3.1. Representative Microleakage Patterns in the Fissurit FX Group

Figure 1 presents a selection of representative samples from the Fissurit FX group, illustrating the range of dye penetration patterns and corresponding scores observed on both surfaces. The visual inspection shows that most sections had minimal to no dye infiltration, while a limited number of samples showed deeper penetration, indicating variability in sealant adaptation.

### 3.2. Representative Dye Penetration Patterns with Perma Seal and Adhesive Application

Representative images of sectioned teeth sealed with Perma Seal combined with an adhesive system, with each sample labeled for buccal (V) and oral (O) microleakage scores are shown into Figure 2. Most samples exhibit complete absence of dye penetration (score 0), and a few isolated samples show minimal leakage (score 1), primarily on one surface only.

These results visually support the quantitative findings, where low and uniform microleakage was observed across both surfaces. The Wilcoxon signed-rank test further confirmed no statistically significant difference between buccal and oral scores, indicating excellent marginal seal performance with the use of an adhesive system.

### 3.3. Representative Dye Penetration Patterns with Perma Seal Without Adhesive Application

This figure displays representative tooth sections treated with Perma Seal without the use of an adhesive system, with each image annotated for buccal (V) and oral (O) microleakage scores. A large proportion of samples exhibit severe dye penetration (score 3) on one or both surfaces, indicating compromised marginal sealing. Only a few samples show minimal or absent leakage (score 0), emphasizing the variability and frequent failure of the material in the absence of a bonding agent. These visual findings are consistent with the quantitative analysis, which revealed the highest average ranks of microleakage in this group and confirmed significant differences compared to the other groups.

### 3.4. Intergroup Comparison of Microleakage Patterns Based on Representative Samples

The comparative visual analysis of representative sectioned samples (Figure 1, Figure 2 and Figure 3) highlights the differences in sealing performance between the tested materials. Samples sealed with Fissurit FX and Perma Seal combined with adhesive system exhibited predominantly low or absent microleakage scores, confirming their effective marginal adaptation. In contrast, the Perma Seal group without adhesives demonstrated frequent and extensive dye penetration, with multiple samples showing maximum scores (3) on both buccal and oral surfaces. These visual observations are in line with the statistical results, reinforcing the importance of using an adhesive system to enhance sealant efficacy.

In Fissurit FX and Perma Seal with adhesive system groups, most scores ranged between 0 and 1, indicating minimal dye penetration and consistent marginal sealing across both surfaces and sections. In contrast, the Perma Seal without adhesive group displayed consistently high scores (2 or 3), particularly evident in both buccal and oral aspects of each tooth, suggesting inadequate sealing and increased microleakage. This structured presentation reinforces the quantitative differences highlighted by the Kruskal–Wallis test and supports the conclusion that the adhesive system significantly improves marginal integrity.

### 3.5. Evaluation of Scoring Consistency Using Cohen’s Kappa

To assess the consistency of microleakage scoring, intra-examiner reliability was evaluated through repeated measurements of all specimens included in the study (n = 72). Each sample was independently scored twice by the same examiner, with a one-week interval between evaluations, under identical optical magnification (7.2×) and lighting conditions, following the Williams and Winters scoring criteria. A Crosstabulation analysis was conducted using SPSS (v.26.0), and Cohen’s kappa coefficient was computed to determine the degree of agreement between the two rounds of evaluation. The resulting kappa value was 0.952, with an asymptotic standard error of 0.033, T = 11.444, and *p* < 0.001, indicating a statistically significant and near-perfect level of agreement. The frequency matrix demonstrated that 42 measures of samples received a score of 0 in both evaluations, 10 measures samples were consistently scored as 1, and 17 measures of samples were consistently scored as 3. Minor discrepancies were noted in only 2 cases, where samples initially classified as Score 2 were later reevaluated as Score 1 or 3. This extremely high concordance confirms the reproducibility and methodological reliability of the examiner’s scoring process.

The microleakage scores (range: 0–3) were treated as ordinal variables, as they represent ranked categories rather than continuous measurements. Although the overall mean score was 0.94 (SD = 1.277), statistical comparisons between groups were based on medians and appropriate non-parametric tests, in accordance with the nature of the data.

### 3.6. Statistical Differences in Microleakage Among Experimental Groups

#### 3.6.1. Power and Sensitivity Analysis

The omnibus Kruskal–Wallis test indicated significant between-group differences (H = 10.243, df = 2, *p* = 0.006). The magnitude of the effect was large (η^2^(H) ≈ 0.55; Cohen’s f ≈ 1.10), implying an achieved power of ~0.99 under the standard one-way ANOVA approximation for n = 18. Individual ordinal microleakage scores by surface (buccal/oral) and protocol are summarized in Table 3 to facilitate side-by-side comparisons.

#### 3.6.2. Pairwise Comparisons Between Groups

Therefore, to evaluate differences in microleakage performance between the three experimental groups—Fissurit FX, Perma Seal with adhesive, and Perma Seal without adhesive—the Kruskal–Wallis H test was applied. A significance level of α = 0.05 was adopted for all analyses. The *p*-value (0.006) is lower than the significance threshold (0.05), therefore we reject the null hypothesis. This indicates that there is a statistically significant difference in microleakage scores between at least two of the groups.

The only statistically significant difference after Bonferroni correction was between Perma Seal with adhesive and Perma Seal without adhesive (*p*-adj = 0.004). The differences between Fissurit FX and the other two groups were not statistically significant after correction (Table 4).

A triangular diagram (Figure 4) illustrates the pairwise average rank comparisons between the three experimental groups as calculated in the Kruskal–Wallis H test. Each vertex represents the mean rank of microleakage scores for a particular group. Group Perma Seal with Adhesive System (AS) had the lowest average rank (28.25), suggesting superior sealing ability and reduced microleakage. Group Fissurit FX followed with an intermediate average rank of 35.92, indicating moderate performance. Group Perma Seal without AS had the highest average rank (45.33), corresponding to poorer sealing quality and greater dye penetration. The spatial arrangement of the nodes confirms that the largest rank difference exists between Perma Seal with adhesive and Perma Seal without adhesive, which aligns with the statistically significant result obtained in post hoc testing (adjusted *p* = 0.004). This visualization supports the conclusion that the presence of an adhesive system significantly reduces microleakage, while Fissurit FX exhibits intermediate behavior.

#### 3.6.3. Buccal vs. Oral Surface Comparison—Fissurit FX Group

A Wilcoxon signed-rank test was applied to compare microleakage scores between the buccal and oral surfaces of the same samples treated with Fissurit FX (n = 12 half-samples per group). The results indicated no statistically significant difference between the two surfaces (Z = 0.184, *p* = 0.854), suggesting a uniform sealing performance of Fissurit FX on both buccal and oral aspects. Test statistics = 5500; Standard error = 2.716; Asymptotic significance (2-tailed) = 0.854. This finding supports the hypothesis that the marginal integrity provided by Fissurit FX is consistent across both surfaces.

#### 3.6.4. Buccal vs. Oral Surface Comparison—Perma Seal with Adhesive System

A Wilcoxon signed-rank test was conducted to compare microleakage scores between the buccal and oral surfaces of the same samples sealed with Perma Seal combined with an adhesive system (n = 12 half-samples per group). The analysis revealed no statistically significant difference between the two surfaces (Z = −0.378, *p* = 0.705), indicating that the sealing performance was consistent across buccal and oral surfaces. Test statistics = 4000; Standard error = 2.646; Asymptotic significance (2-tailed) = 0.705. These results suggest that the use of an adhesive system contributes to a uniform marginal seal, independent of surface orientation.

#### 3.6.5. Buccal vs. Oral Surface Comparison—Perma Seal Without Adhesive System

A Wilcoxon signed-rank test was also performed for the group sealed with Perma Seal without adhesive system (n = 12 half-samples per group). The results showed no statistically significant difference in microleakage scores between the buccal and oral surfaces (Z = 0.447, *p* = 0.655), indicating that, despite higher overall leakage, the material behaved similarly on both surfaces. Test statistics = 2000; Standard error = 1.118; Asymptotic significance (2-tailed) = 0.655.

None of the tested materials showed significant differences in microleakage scores between buccal and oral surfaces. This suggests that surface orientation does not influence the sealing performance of the applied materials, regardless of the presence or absence of an adhesive system. Wilcoxon signed-rank tests were conducted for each sealant group to compare microleakage scores between buccal and oral surfaces of the same tooth sections. None of the groups showed statistically significant differences, indicating uniform sealing performance across surfaces. The distribution of paired differences between buccal and oral microleakage scores for each material group is presented in Figure 5a–c. All three groups demonstrated no statistically significant differences (Wilcoxon signed-rank test), supporting the visual impression of surface-independent sealant behavior.

#### 3.6.6. Chi-Square Analysis of Microleakage Scores by Sealant Type

The association between the type of sealant material and microleakage score was assessed using the Chi-Square Test of Independence. The test revealed a statistically significant relationship between sealant group and score category (χ^2^ = 20.219, df = 6, *p* = 0.003). This indicates that the distribution of microleakage scores varied significantly depending on the sealant used. A total of 72 observations were analyzed. The highest frequency score of 0 (no dye penetration) was observed in the Perma Seal + adhesive group, while higher scores (2 and 3) were more common in the Perma Seal without adhesive group. The Fissurit FX group showed an intermediate performance. Although 3 cells (50.0%) had expected counts less than 5, which may affect the precision of the test, the association remained statistically significant. This supports the visual and descriptive findings, suggesting that the presence of an adhesive system significantly influences the sealing capacity of the materials tested.

## 4. Discussion

The present in vitro study evaluated and compared the microleakage performance of three commonly used pit-and-fissure sealant protocols—Fissurit FX, Perma Seal with adhesive system, and Perma Seal without adhesive—under standardized laboratory conditions. The use of dye penetration scoring and non-parametric statistical tests allowed for the assessment of sealing ability and marginal integrity, which are critical for the long-term clinical success of sealants. This investigation contributes to the growing body of literature supporting the use of adhesive systems to enhance sealant performance in preventive dentistry.

The Kruskal–Wallis test showed a statistically significant difference in microleakage scores among the three groups (*p* = 0.006). Perma Seal, used with an adhesive system, demonstrated the lowest microleakage scores, while Perma Seal without adhesive showed the highest leakage values. These findings align with recent in vitro studies which demonstrated that reapplication of acid-etching and adhesive pretreatment after saliva contamination significantly reduced microleakage in fissure sealants [11]. This evidence supports the principle that bonding agents facilitate improved resin infiltration and marginal seal.

Similarly, other authors found that the use of bonding agents, especially staged-curing protocols or application after enamel contamination, significantly decreased microleakage in permanent molars [5,12]. The current study corroborates that bonding agents enhance micromechanical retention and protect enamel microporosities from leakage.

The Wilcoxon signed-rank test indicated no significant differences in microleakage between buccal and oral halves within each group, implying consistent application technique. Visual inspection of sectioned specimens supported this, showing extensive dye penetration in non-adhesive groups and minimal infiltration in adhesive-treated groups—paralleling the findings of Jalannavar & Rajguru (2024), who stressed the importance of surface conditioning and operator consistency in achieving uniform sealant application [13].

Chi-Square analysis demonstrated a statistically significant association between sealant material type and microleakage score (χ^2^ = 20.219, df = 6, *p* = 0.003). Consistent with these findings, Almahdy et al. (2021) reported that using a bonding agent prior to fissure sealant application significantly reduced microleakage and enhanced micro tensile bond strength, particularly under aging conditions [14]. These results further support the clinical recommendation to incorporate adhesive pretreatment protocols when applying sealants, to optimize marginal adaptation and minimize leakage, especially in fissured enamel anatomy.

High intra-examiner agreement was confirmed through Cohen’s kappa coefficient (κ = 0.952). The high reproducibility achieved in this study reinforces the reliability of the method used for microleakage evaluation and supports the validity of statistical outcomes.

Clinically, these findings support the routine use of an adhesive system prior to sealant application, particularly when using low-viscosity resin sealants. Although Fissurit FX provided acceptable sealing, the consistency of low microleakage scores was superior in the Perma Seal + adhesive group. This may be attributed to differences in filler content and resin composition, as suggested by studies such as those of Deery (2013), which underline the impact of material properties on sealant retention and performance [15].

A systematic review and meta-analysis by Fumes et al. (2017) concluded that preconditioning enamel with phosphoric acid results in significantly lower microleakage in fissure sealants compared to Er:YAG laser or air-abrasion methods (*p* < 0.001), reinforcing the procedural importance of acid etching in sealant protocols [16]. One study analyzing the composition and surface morphology of bioadhesive GICs demonstrated that the microstructural uniformity and filler distribution play a critical role in bonding performance and interface stability, which may partially explain their variable clinical outcomes compared to resin-based sealants [17].

The histological complexity of the tooth—particularly the interface between enamel, dentin, and pulp—plays a critical role in the success of pit-and-fissure sealants. Enamel, being an acellular and highly mineralized tissue, offers a rigid but non-regenerative substrate, making the integrity of the initial seal essential [18]. Beneath the enamel, dentin’s tubular structure and vital nature introduce the risk of bacterial infiltration if microleakage occurs [8]. Thus, any compromise in marginal adaptation of the sealant may lead to dentin exposure and subsequent pulpal irritation. Understanding the layered histological architecture of the tooth underscores the importance of achieving a tight, continuous seal along fissures, particularly in deep anatomical grooves where enamel is thinner and irregular, increasing susceptibility to leakage and carious progression [19]. The results of the present study are consistent with those reported by Gorseta et al. [20], who performed an in vitro analysis comparing the microleakage of self-adhesive fissure sealants with that of conventional resin-based and glass ionomer cement (GIC) sealants. Their findings demonstrated significantly lower microleakage in the conventional resin-based sealant group, while the self-adhesive sealants exhibited intermediate values, and GIC-based sealants showed the highest degree of leakage. These observations reinforce the notion that although self-adhesive sealants simplify the clinical procedure, their sealing capacity may be compromised when not used in conjunction with an adhesive system. This supports our findings, which showed superior marginal sealing in the group using an adhesive pretreatment prior to sealant placement. As the materials are based on hydrophobic resin systems, they are very sensitive to moisture contamination which is a significant issue in young children and partially erupted teeth [21]. An in vivo study reported statistically significant differences in color stability and postoperative sensitivity between two hybrid resin composites used in minimally invasive restorations [22]. These clinical outcomes align with our in vitro findings, supporting the importance of adhesive systems and precise material adaptation in minimizing microleakage and enhancing marginal seal integrity. From a clinical perspective, the systematic review by Memarpour et al. (2023) demonstrated that enamel contamination significantly reduces both the shear bond strength and sealing ability of fissure sealants, highlighting the critical importance of maintaining proper isolation during sealant application to ensure long-term success [23]. Also from a clinical standpoint, the systematic review by Ferreira et al. (2023) identified multiple factors influencing the shear bond strength of pit and fissure sealants—such as enamel surface treatment, sealant viscosity, and curing technique—which are critical determinants of long-term retention and marginal sealing performance [24]. The results of an in vitro and in vivo study have highlighted the importance of resin composite microstructure in ensuring long-term sealant retention. Using atomic force microscopy, the sealant material—based on bis-GMA—was shown to possess a homogeneous surface without structural discontinuities, which likely contributed to its favorable marginal adaptation. Clinical evaluation confirmed that this structural integrity translated into a high level of retention and sealing performance. These findings support the current evidence that optimized material composition, particularly in terms of filler distribution and resin matrix stability, plays a significant role in minimizing microleakage and enhancing clinical durability [25]. A recent study evaluated the physicochemical properties and anti-biofilm potential of technological dental sealants, confirming that material innovations can significantly enhance both the sealing efficacy and resistance to biofilm formation in vitro [26]. An in vitro study by Juntavee et al. (2023) demonstrated that both the type of sealant and the application technique significantly influence microleakage and fissure penetration, highlighting the importance of protocol standardization to optimize sealant efficacy [27]. Dixit et al. (2021) showed that the penetration depth and microleakage of pit and fissure sealants vary significantly depending on the material used, emphasizing the importance of selecting sealants with superior flow properties to improve marginal sealing and long-term effectiveness [28]. Although Fissurit FX contains sodium fluoride (NaF ≤ 2.5%) and offers potential benefits through the localized release of fluoride ions, it is important to distinguish between intrinsic fluoride release from the material itself and prior topical fluoride treatments. According to the systematic review by Teimoory et al. (2023) [29], the application of topical fluoride before sealant placement can negatively affect the bond strength, likely due to the formation of a hyper mineralized enamel surface that hinders resin infiltration. Therefore, while fluoride-releasing sealants like Fissurit FX may enhance caries resistance, which must be taken not to combine them with immediate pre-treatment using topical fluorides, as this may compromise adhesive performance [29].

From a restorative perspective, the findings of Tavangar et al. (2021) [30] emphasize the importance of surface treatment in enhancing the shear bond strength between new and aged composite materials. Although their study focused on composite–composite repair, the principles of interfacial adhesion and surface conditioning are equally relevant to pit-and-fissure sealant adhesion, particularly in cases of sealant reapplication or repair, where achieving optimal micromechanical retention is crucial for clinical longevity [30].

Although not directly related to sealant application, the study by Marín-Velasquez et al. (2024) highlights how finishing protocols influence the surface quality of nanohybrid composites, underscoring the broader relevance of surface treatment techniques in maintaining the functional and aesthetic integrity of resin-based materials [31].

Although not directly related to fissure sealant application, the study by Soliman et al. (2021) [32] demonstrated that the choice of polishing system significantly affects surface roughness and gloss in nanohybrid composites. These findings reinforce the importance of surface finishing protocols in optimizing the performance and longevity of resin-based materials used in preventive and restorative dentistry [32].

Similarly, Alharbi et al. (2024) [33] investigated various polishing systems on universal single-shade composites and confirmed that surface characteristics are highly influenced by the post-application treatment approach. This has clinical implications not only for aesthetics but also for minimizing plaque retention and enhancing marginal adaptation—considerations that are equally important in sealant performance [33].

### Limitations

The relatively small sample size (n = 6 per group) represents a limitation of this study, although it is in line with other in vitro microleakage investigations and consistent with ISO/TS 11405:2015 recommendations for adhesion testing. In addition, the thermocycling protocol of 500 cycles reflects the minimal standard for artificial aging; however, extended protocols with higher numbers of cycles and additional aging methods would better simulate long-term clinical conditions. Future studies with larger samples and more extensive aging protocols are therefore needed to confirm and expand upon these findings. Another limitation of this study is the composition of the sample, which included 15 permanent molars and 3 permanent premolars. To the present in vitro investigation, all specimens were considered together as posterior teeth, since the evaluation focused on the occlusal enamel surface. However, we acknowledge that structural differences between molars and premolars may affect sealing performance, and future studies with larger and separately analyzed groups are required to further explore this aspect.

Future studies should include larger sample sizes and explore the use of advanced imaging techniques such as scanning electron microscopy (SEM) or micro-CT for detailed marginal analysis. In vivo studies comparing adhesive protocols under clinical conditions it is also necessary to confirm the long-term benefits observed in vitro.

## 5. Conclusions

Within the limitations of this in vitro study, our findings emphasize the important role of adhesive systems in improving the marginal integrity of pit-and-fissure sealants. The application of an adhesive layer before sealant placement enhances marginal sealing, suggesting that this additional step may increase the long-term effectiveness of preventive sealing therapy.

From a clinical perspective, incorporating an adhesive system may be particularly valuable in pediatric dentistry, where moisture control and the complex occlusal morphology of molars often challenge the durability of sealants. Nevertheless, further studies with larger samples, extended artificial aging protocols, and clinical trials are needed to validate these findings and provide stronger evidence for daily practice.

## Figures and Tables

**Figure 1 biomedicines-13-02902-f001:**
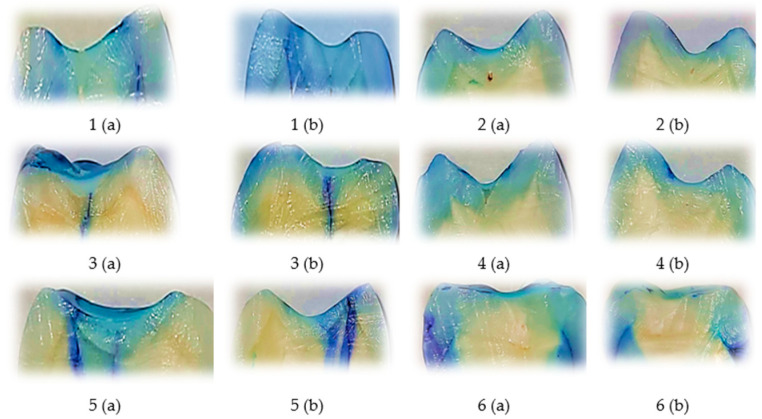
Representative longitudinally sectioned tooth samples treated with Fissurit FX (VOCO GmbH, Cuxhaven, Germany), showing dye penetration and microleakage patterns. Each tooth was sectioned into two halves, indicated as (a) and (b). Buccal (V) and oral (O) microleakage scores were recorded as follows: 1 (a): V = 1, O = 1; 1 (b): V = 0, O = 0; 2 (a): V = 0, O = 0; 2 (b): V = 0, O = 0; 3 (a): V = 3, O = 3; 3 (b): V = 3, O = 3; 4 (a): V = 1, O = 0; 4 (b): V = 1, O = 0; 5 (a): V = 3, O = 0; 5 (b): V = 2, O = 0; 6 (a): V = 0, O = 0; 6 (b): V = 0, O = 0. Scale bar = 1 mm.

**Figure 2 biomedicines-13-02902-f002:**
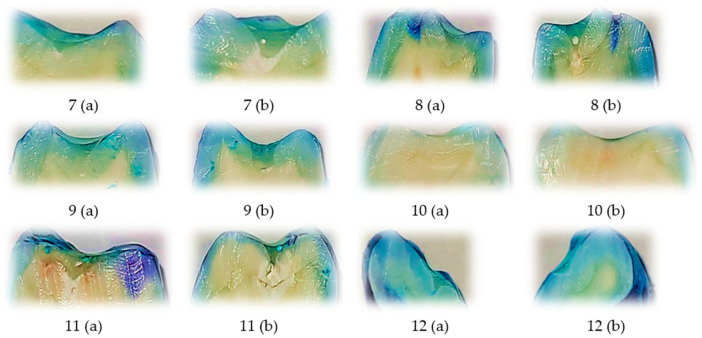
Representative longitudinally sectioned tooth samples treated with Perma Seal (Ultradent Products Inc., South Jordan, UT, USA) combined with an adhesive system, showing dye penetration and microleakage patterns. Each tooth was sectioned into two halves, indicated as (a) and (b). Buccal (B) and oral (O) microleakage scores were recorded as follows: 7 (a): B = 0, O = 0; 7 (b): B = 0, O = 0; 8 (a): B = 0, O = 0; 8 (b): B = 0, O = 0; 9 (a): B = 1, O = 0; 9 (b): B = 1, O = 1; 10 (a): B = 0, O = 0; 10 (b): B = 0, O = 0; 11 (a): B = 0, O = 2; 11 (b): B = 1, O = 0; 12 (a): B = 1, O = 0; 12 (b): B = 0, O = 0. Scale bar = 1 mm.

**Figure 3 biomedicines-13-02902-f003:**
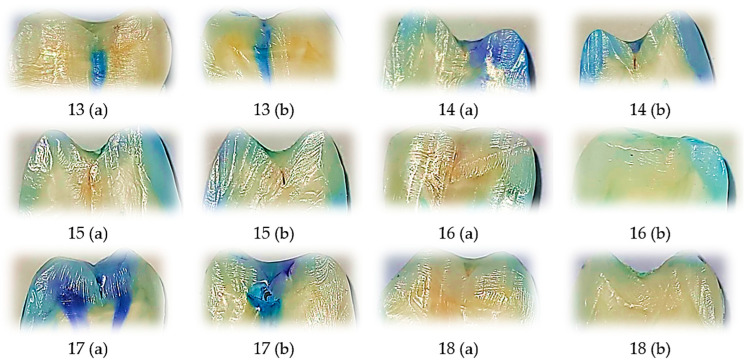
Representative longitudinally sectioned tooth samples treated with Perma Seal (Ultradent Products Inc., South Jordan, UT, USA) without an adhesive system, showing dye penetration and microleakage patterns. Each tooth was sectioned into two halves, indicated as (a) and (b). Buccal (B) and oral (O) microleakage scores were recorded as follows: 13 (a): B = 3, O = 3; 13 (b): B = 3, O = 3; 14 (a): B = 3, O = 3; 14 (b): B = 3, O = 3; 15 (a): B = 3, O = 0; 15 (b): B = 1, O = 0; 16 (a): B = 0, O = 0; 16 (b): B = 0, O = 0; 17 (a): B = 3, O = 3; 17 (b): B = 3, O = 3; 18 (a): B = 0, O = 0; 18 (b): B = 0, O = 0. Scale bar = 1 mm.

**Figure 4 biomedicines-13-02902-f004:**
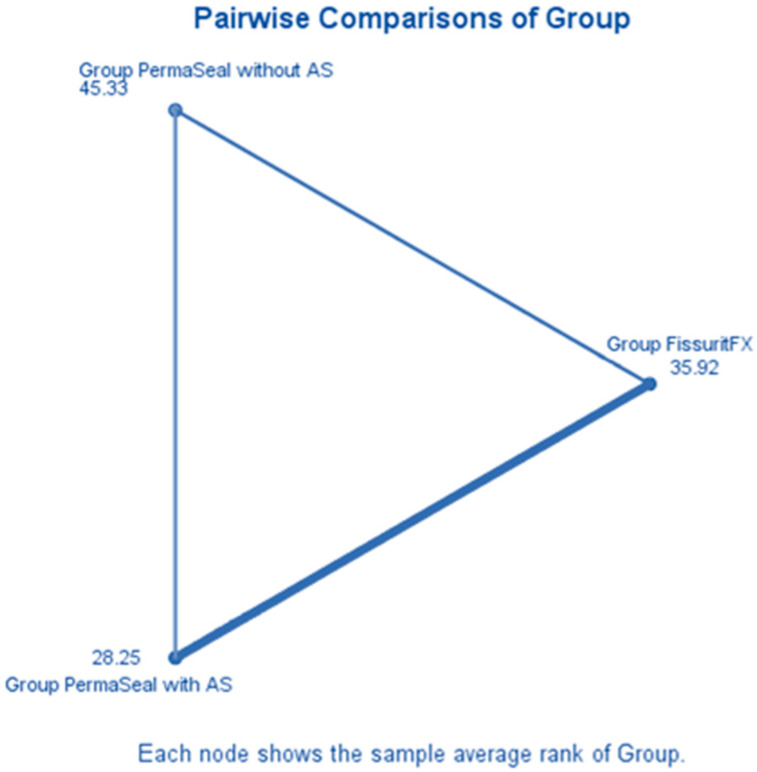
Mean rank differences in microleakage scores among the experimental groups (Kruskal–Wallis test).

**Figure 5 biomedicines-13-02902-f005:**
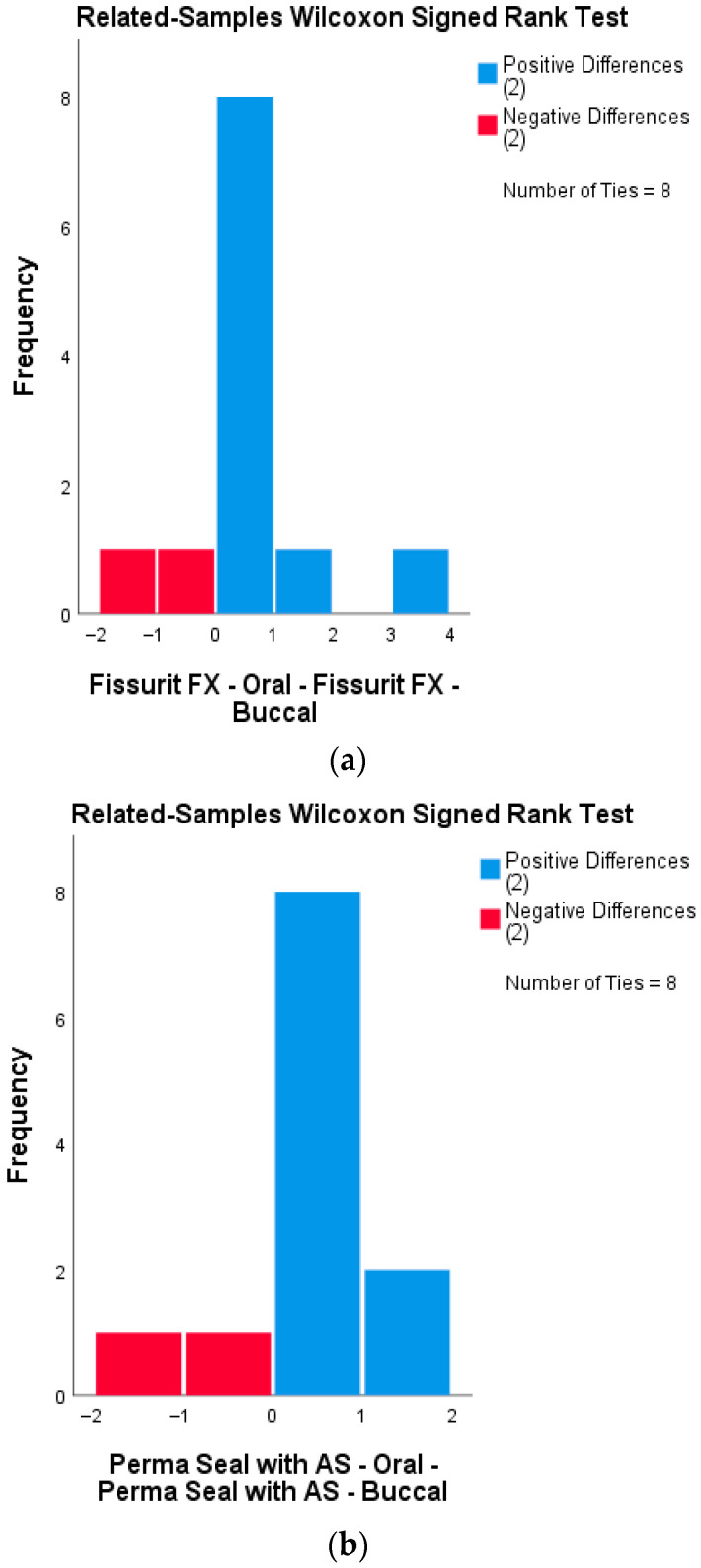
Distribution of differences between buccal and oral microleakage scores, analyzed using the Wilcoxon signed-rank test for each sealant group. (**a**) Fissurit FX group: Distribution of signed differences between buccal and oral surfaces. Most differences are zero or minimal, indicating consistent performance across surfaces. (**b**) Perma Seal with adhesive system: All matched pairs showed no or minimal difference in microleakage scores, confirming high uniformity of seal integrity. (**c**) Perma Seal without adhesive system: Although most differences remain low, several samples show discrepancies in microleakage between surfaces.

**Table 1 biomedicines-13-02902-t001:** Material Manufacturer Composition Filler (%) Lot No.

Material	Manufacturer	Composition	Filler (%)	Lot
Fissurit FX	VOCO GmbH, Germany	TEGDMA (10–25%), UDMA (10–25%), BIS-GMA (5–10%), BIS-EMA (5–10%), NaF ≤ 2.5%	55%	2201408
PermaSeal	Ultradent, USA	TEGDMA ≤ 25%, BIS-GMA 10%, MFP 0.3%	68%	1-800-552-5512
Xbond adhesive	Schulzer, Waghäusel, Germany	One-step light-cured bonding agent (no filler)	–	52104917D
Etchgel	Itena Clinical, Villepinte, France	36% orthophosphoric acid gel	–	1639

**Table 2 biomedicines-13-02902-t002:** Sealant Application Protocol Application steps were standardized across groups.

Protocol Step	Group 1 Fissurit FX + Xbond	Group 2 Perma Seal + Xbond	Group 3 Perma Seal (No Adhesive)
Etching with 36% phosphoric acid gel	Yes	Yes	Yes
Rinsing and drying (chalky white enamel)	Yes	Yes	Yes
Application of Xbond adhesive	Yes	Yes	No
Air-drying adhesive (5 s)	Yes	Yes	No
Light-curing adhesive (20 s)	Yes	Yes	No
Application of sealant	Yes	Yes	Yes
Sealant type	Fissurit FX	PermaSeal	PermaSeal
Light-curing sealant (20 s)	Yes	Yes	Yes
Use of bonding agent	Yes	Yes	No
Dye immersion and microleakage evaluation	Yes	Yes	Yes

**Table 3 biomedicines-13-02902-t003:** Individual microleakage scores (0–3) on buccal and oral surfaces by sealant protocol. (a) and (b) denote the two halves of each tooth.

	Score							
	Surface			Surface			Surface	
	B	O		B	O		B	O
F-FX			PS + AS			PS-AS		
1 (a)	1	1	7 (a)	0	0	13 (a)	3	3
1 (b)	0	0	7 (b)	0	0	13 (b)	3	3
2 (a)	0	0	8 (a)	0	0	14 (a)	3	3
2 (b)	0	0	8 (b)	0	0	14 (b)	3	3
3 (a)	3	3	9 (a)	1	0	15 (a)	3	0
3 (b)	3	3	9 (b)	1	1	15 (b)	1	0
4 (a)	1	0	10 (a)	0	0	16 (a)	0	0
4 (b)	1	0	10 (b)	0	0	16 (b)	0	0
5 (a)	3	0	11 (a)	0	2	17 (a)	3	3
5 (b)	2	0	11 (b)	1	0	17 (b)	3	3
6 (a)	0	0	12 (a)	1	0	18 (a)	0	0
6 (b)	0	0	12 (b)	0	0	18 (b)	0	0

Abbreviations: B = Buccal; O = Oral; F-FX = Fissurit FX; PS + AS = Perma Seal with adhesive system; PS-AS = Perma Seal without adhesive system. Note: n = 6 teeth per group (18 total).

**Table 4 biomedicines-13-02902-t004:** Pairwise Comparisons between Groups (Post hoc Test with Bonferroni Correction).

Pairwise Comparisons of Group					
Sample 1–Sample 2	Test Statistic	Std. Error	Std. Test Statistic z-Score	*p*-Value	Adjusted *p*-Value ^a^ (Bonferroni)
Group Perma Seal with AS-Group Fissurit FX	7.667	5.347	1.434	0.152	0.455
Group Perma Seal with AS-Group Perma Seal without AS	−17.083	5.347	−3.195	0.001	0.004
Group Fissurit FX-Group Perma Seal without AS	−9.417	5.347	−1.761	0.078	0.235

Each row tests the null hypothesis that the Sample 1 and Sample 2 distributions are the same. Asymptotic significances (2-sided tests) are displayed. The significance level is 0.05. ^a^ Significance values have been adjusted by the Bonferroni correction for multiple tests.

## Data Availability

We share the research data processing in three sections.

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
