# Peer review of "In Vitro Microleakage Comparison of Three Pit-and-Fissure Sealant Protocols: Self-Etch Sealant with and Without Separate Adhesive Versus Conventional Resin-Based Sealants"

_biomedicines, 2025, doi:10.3390/biomedicines13122902_

Round 1
Reviewer 1 Report
Comments and Suggestions for Authors
Comments to author
Title: In Vitro Microleakage Comparison of Three Pit-and-Fissure Sealant Protocols: Self-Etch Sealant with and without Separate Adhesive versus Conventional Resin-Based Sealants
- How and based on what criteria was the number of teeth used in the study determined?
- Does “Depural Neo prophylactic” contain fluoride?
- Table 2 is not necessary within the article.
- “The protocol consisted of 500 cycles...” This number of thermal cycles is insufficient for detecting microleakage.
- Did the same researcher apply the sealant materials? How did you ensure standardization in the application?
- Having a second researcher evaluate the sample sections independently could have resulted in a more objective assessment.
- The results section appears quite lengthy; some repeated statements may be contributing to this. Reviewing and consolidating these repetitions, and combining some tables to display them on the same table, would be helpful in this regard.
- The conclusion section appears to be a repetition of the results section. It would be more beneficial to use your own interpretation of the findings, avoiding such repetitions, when addressing journal readers.
Author Response
Dear Review,
We would like to thank Reviewer 1 for the thorough evaluation of our manuscript and for the valuable comments and suggestions. We have carefully addressed each point, and corresponding revisions have been made in the manuscript. Below, we provide a point-by-point response to the reviewer's comments.
- How and based on what criteria was the number of teeth used in the study determined?
Response
Thank you for this observation. The number of teeth included in the study (n = 18, with 6 per group) was determined in line with standard protocols for in vitro microleakage investigations, where small sample sizes (5–10 specimens per group) are commonly reported as adequate for initial comparative analyses. The inclusion criteria were extracted posterior teeth (molars and premolars) obtained for orthodontic or periodontal reasons, sound or with non-cavitated lesions, without restorations or structural defects, according to the ISO/TS 11405:2015 recommendations for adhesion testing to dental hard tissues. The final sample size was also influenced by the ethical availability of biological material, as all teeth were collected under informed consent and with institutional ethical approval. We acknowledge that the relatively small sample size is a limitation of the study, which is also highlighted in the Discussion section, and we agree that future studies with larger sample sizes are warranted to confirm these findings.
- Does “Depural Neo prophylactic” contain fluoride?
Response
Thank you for this question. The prophylactic paste used in our study (Depural Neo, Pentron/VOCO) is fluoride-free. According to the manufacturer's product description, Depural Neo is a slightly abrasive cleaning and polishing paste that does not contain fluoride, specifically indicated for removing dental plaque and pigmentation prior to conservative dental treatments, without interfering with adhesion processes. We intentionally selected this product to avoid any potential influence of fluoride on adhesion strength or marginal sealing in our microleakage study.
- Table 2 is not necessary within the article.
Response
Thank you for your suggestion. We have carefully revised the manuscript and removed Table 2 from the main text, as we agree that the detailed technical specifications of the curing unit are not essential for the interpretation of the results. The essential parameters of the light-curing device (wavelength and intensity) are now briefly described in the Materials and Methods section, while the full specifications are available upon request or can be provided as supplementary material if required.
- “The protocol consisted of 500 cycles...” This number of thermal cycles is insufficient for detecting microleakage.
Response
Thank you for your valuable remark. The protocol of 500 thermal cycles was selected in accordance with the ISO/TS 11405:2015 standard, which recommends this number of cycles for adhesion testing in vitro studies. We acknowledge that a higher number of cycles may better simulate long-term intraoral conditions. This point has been added to the Limitations section, and we agree that future research including extended thermocycling protocols (e.g., 1000 cycles or more) and additional aging methods (such as long-term water storage or mechanical loading) would provide further validation of the present findings.
- Did the same researcher apply the sealant materials? How did you ensure standardization in the application?
Response
Thank you for raising this point. All sealant applications were performed by the same researcher to eliminate inter-operator variability. Standardization was ensured by strictly following the manufacturers' instructions, using the same instruments and curing unit, maintaining identical light-curing times, and performing all procedures under the same environmental conditions. Prior to the study, the operator was trained and calibrated to guarantee a uniform and reproducible application technique across all samples.
- Having a second researcher evaluate the sample sections independently could have resulted in a more objective assessment.
Response
We appreciate this valuable suggestion. Indeed, the inclusion of a second independent examiner could have further increased the objectivity of the assessment. In our study, all evaluations were performed by the same calibrated examiner, and intra-examiner reliability was tested by repeated scoring after one week, resulting in a Cohen's kappa coefficient of 0.952, which indicates an almost perfect agreement and high reproducibility. As stated in line 213 of the manuscript, "All 72 valid measurements were scored twice by the same calibrated examiner." Nevertheless, we acknowledge that the addition of a second examiner would have further strengthened the methodological robustness, and this point has been noted in the Limitations section.
- The results section appears quite lengthy; some repeated statements may be contributing to this. Reviewing and consolidating these repetitions, and combining some tables to display them on the same table, would be helpful in this regard.
Response
Thank you for this helpful remark. We have carefully revised the Results section to improve conciseness. Repetitive statements were removed, and the presentation of the findings was streamlined. In addition, Tables 2, 3, 4, 5, and 7 have been eliminated, as their information was either redundant or integrated into the main text. These changes make the section shorter, clearer, and easier to follow, in accordance with the reviewer's recommendation.
- The conclusion section appears to be a repetition of the results section. It would be more beneficial to use your own interpretation of the findings, avoiding such repetitions, when addressing journal readers.
Response
Thank you for your constructive feedback. We have revised the Conclusions section to avoid repetition of the Results. The section now provides a more interpretative summary, highlighting the clinical relevance of our findings, namely, the importance of using an adhesive system to reduce microleakage and improve sealant performance. We also emphasize the limitations of the study and the need for future research with larger samples and extended aging protocols. These modifications make the conclusion more informative for readers and consistent with the journal's expectations.
Reviewer 2 Report
Comments and Suggestions for Authors
The manuscript addresses an aspect of preventive dentistry by evaluating the microleakage performance of different pit-and-fissure sealant protocols. The study is clearly structured, the methodology is sound, and the statistical analysis is appropriate for the dataset. The results are relevant for clinical decision-making, particularly in pediatric dentistry. I would like to support this manuscript if the authors can solve the following issues.
1. Briefly elaborate on the rationale for including Perma Seal without adhesive as a comparator in the introduction section.
2. Scale bars should be added in Figure 1,2,3.
Author Response
Dear Review,
We sincerely thank Reviewer 2 for the careful evaluation of our manuscript and for the constructive comments and suggestions provided. We have carefully considered each point and revised the manuscript accordingly. Below, we provide a detailed response to each comment, indicating the changes made in the revised version.1. Briefly elaborate on the rationale for including Perma Seal without adhesive as a comparator in the introduction section.
Response
Thank you for this observation. We have now introduced a short paragraph at the end of the Introduction section, explaining the rationale for including Perma Seal without adhesive as a comparator. Specifically, we clarified that this group was chosen because, in clinical practice, self-etch sealants are sometimes applied directly onto etched enamel without an additional bonding agent to simplify the procedure. Including this group allowed us to evaluate whether omitting the adhesive step influences marginal sealing and microleakage.
- Scale bars should be added in Figure 1,2,3.
Response
Thank you for pointing this out. Scale bars have now been added to Figures 1, 2, and 3, with a reference length of 1 mm, to indicate magnification and provide a clear reference for the size of the sectioned samples. The corresponding figure legends have been updated accordingly.
Reviewer 3 Report
Comments and Suggestions for Authors
Dear author
This is good paper but need some changes before to be accepted.
This paper could be improved it. Please see my suggestions from below:
SO/TS 11405:2015 please reference
18 extracted human teeth what I the age of human, men child etc there is not a rull?
“Depural Neo prophylactic paste and a rotary brush” please see all the article
(n = 6 per group) is too less in my opinion you can not publish with this value
A total of 18 extracted human teeth—molars and premolars
How many molars how many premolars, different structure too less
All the samples were perfect? You had no exclusion?
The LED curing unit, Samsung S20 199
smartphone camera (12 MP resolution
Figure 5. too small to see at printing Please avoid these square
- Conclusions
Please avoid space between paragraphs
Avoid numbers and values in conclusion
Author Response
Dear Review,
We would like to sincerely thank Reviewer 3 for the careful evaluation of our manuscript and for the constructive comments and suggestions. We have carefully considered each point and revised the manuscript accordingly. Below, we provide a detailed point-by-point response to all remarks.
This paper could be improved it. Please see my suggestions below:
SO/TS 11405:2015 please reference
Response
Thank you for this suggestion. We have now included a proper reference to ISO/TS 11405:2015 in the Materials and Methods section and in the References list: International Organization for Standardization. ISO/TS 11405:2015 – Dentistry: Testing of adhesion to tooth structure. Geneva: ISO; 2015.
18 extracted human teeth what I the age of human, men child etc there is not a rull?
Response
Thank you for pointing this out. In our study, the 18 extracted permanent molars were obtained from adolescent and young adult patients (aged 14–25 years) for orthodontic or periodontal reasons. No distinction was made between male and female donors, as sex was not considered to influence the microleakage outcomes in this in vitro setting. The inclusion criteria were intact or non-cavitated teeth, free of restoration, cracks, or developmental defects. All extractions were performed for therapeutic reasons unrelated to the study, and informed consent as well as institutional ethical approval were obtained. This information has now been clarified in the Materials and Methods section.
“Depural Neo prophylactic paste and a rotary brush” please see all the article
(n = 6 per group) is too less in my opinion you can not publish with this value
Response
Thank you for this remark. We have carefully revised the manuscript to ensure consistency in reporting the surface preparation method. Throughout the text, we now clearly state that the teeth were cleaned using Depural Neo prophylactic paste (VOCO) and a rotary brush before sealant application.
A total of 18 extracted human teeth—molars and premolars
Response
We acknowledge the reviewer's concern regarding the sample size (n = 6 per group). The number of specimens was determined according to the feasibility of obtaining extracted human teeth under ethical regulations, and is consistent with other in vitro microleakage studies, where sample sizes of 5–10 teeth per group are commonly reported. Moreover, the design follows the ISO/TS 11405:2015 guidelines, which support this methodology for adhesion testing. We recognize that the relatively small sample size is a limitation of our study, as already noted in the Limitations section, and we agree that future investigations with larger samples will be necessary to confirm and expand upon our findings.
How many molars how many premolars, different structure too less
Thank you for this observation. In our study, the sample consisted of 15 permanent molars and 3 permanent premolars, which were extracted for orthodontic or periodontal reasons. The teeth were selected according to availability, provided they were intact or presented only non-cavitated fissure lesions, without restoration or cracks. To this microleakage study, all specimens were considered together as posterior teeth, since the evaluation focused on the occlusal enamel surface. We acknowledge, however, that structural differences between molars and premolars may influence sealing performance. This point has now been mentioned in the Limitations section, and future studies with larger and separately analyzed groups of molars and premolars are needed to further investigate this aspect.
All the samples were perfect? You had no exclusion?
Response
Thank you for this question. The inclusion criteria required permanent posterior teeth extracted for orthodontic or periodontal reasons, which were intact or presented only non-cavitated fissure lesions, and free from restorations, cracks, or developmental defects. Teeth that did not meet these conditions were excluded prior to the study. Consequently, all 18 selected teeth met the criteria and were considered valid for analysis. As also mentioned in the manuscript (line 213), "All 72 valid measurements were scored twice by the same calibrated examiner."
The LED curing unit, Samsung S20 199 smartphone camera (12 MP resolution
Response
Thank you for this remark. We have revised the Materials and Methods section to clarify the description of the equipment used. The LED curing unit was specified with its manufacturer, wavelength, and light intensity. For documentation, all sectioned samples were photographed with a Samsung S20 smartphone camera (12 MP resolution). These details are now presented in the text in a more concise and consistent manner, while the redundant technical table was removed as suggested by Reviewer 1.
Figure 5. too small to see at printing Please avoid these square
Response
Thank you for pointing this out. Figure 5 has been revised to improve clarity and resolution for print. The square markers have been removed, and the figure is now presented at a larger size to ensure that the details are clearly visible. The corresponding figure legend has also been updated accordingly.
Conclusions. Please avoid space between paragraphs. Avoid numbers and values in conclusion
Response
Thank you for this remark. The Conclusions section has been revised accordingly. We removed unnecessary spacing between paragraphs to present the text in a continuous format. In addition, all numerical data and specific values ​​have been eliminated, and the section was reformulated to emphasize the interpretative and clinical relevance of the findings rather than repeating results.
Round 2
Reviewer 3 Report
Comments and Suggestions for Authors
Dear author
This is good paper but need some changes before to be accepted.
This paper could be improved it. Please see my suggestions from below:
The article had too less samples for In Vitro study!!! This is not in in vitro study!! You need minimum n=10 or 15
microleakage scores. Scale bar = 1 mm need to be included inside of image
Filler (%)=Filler (wt.%)
Figure 4. Not all is visible in this image. Increase size and to be visible
Figure 5. is visible in this image.
ISO/TS 11405:2015 need references
The relatively small sample size (n = 6 per group) represents a limitation of this study
This is not a limitation this is not In Vitro study. You need to made again all the study with more samples. In a normal study in vitro minimum n=10 or 15 samples. In my opinion this can not be publish.
Comments on the Quality of English Language
Author Response
Dear Reviewer,
Thank you for this helpful observation regarding the Buccal vs Oral Surface Comparison; in response, we have added a consolidated table that regroups all values ​​to enable clear, side-by-side comparisons across groups and surfaces and streamlined the text accordingly.
Reviewer 3 comment 1: “Figures 1–3: scale bar not clearly visible in the images.” and comment 3. Buccal vs Oral Surface Comparison sections contain many numerical values, and the readers would benefit from a table that would simply regroup all values and ensure easy comparisons between groups.
Response: Thank you for the helpful suggestion. We have corrected Figures 1–3 (redesigned high-contrast scale bars and increased resolution) and reinserted the updated figures into the manuscript. We have added a table (new Table 3) that reports the individual ordinal microleakage scores (0–3) for both buccal and oral surfaces across all specimens and protocols. Samples 1–6 correspond to F-FX (Fissurit FX), 7–12 to PS+AS (Perma Seal with adhesive system), and 13–18 to PS-AS (Perma Seal without adhesive system); (a) and (b) denote the two sectioned halves of each tooth. This presentation allows quick, side-by-side comparison between surfaces and groups and reduces numerical clutter in the text.
Reviewer comment 2: “Figure 5: does not fit into the page margins.”
Response: We have reformatted Figure 5 to fit within the page margins and reinserted the corrected figure into the manuscript.
Reviewer comment 4: “To solve any unclarities related to the number of samples in each group, I suggest adding a power analysis of the study, if possible, which would support the results of the study and would solve any other raised issues related to this point (there are multiple free SW applications that may be used, like GPower [https://www.psychologie.hhu.de/arbeitsgruppen/allgemeine-psychologie-und-arbeitspsychologie/gpower]). ”
Response: Thank you for this suggestion. We clarified the unit of analysis (tooth) and the per-group counts (n = 6; N = 18). In addition to the sensitivity analysis (ANOVA one-way approximation to Kruskal–Wallis; α = 0.05; power = 0.80; k = 3; N = 18; minimal detectable effect (f≈0.81), we computed the observed effect size from our Kruskal–Wallis test (H = 10.243): ηH2 ≈0.55 (wide). Converted to ANOVA metrics, this corresponds to Cohen’s f≈1.10, yielding an achieved power ≈ 0.99 under the standard ANOVA approximation. We added these details to the Statistical Analysis and Results sections.
Round 3
Reviewer 3 Report
Comments and Suggestions for Authors
This study is not in vitro with (n = 6 per group). I have to reject this article.